# Amassing the Security: An Enhanced Authentication Protocol for Drone Communications over 5G Networks

**Tsuyang Wu** [1], **Xinglan Guo** [1], **Yehcheng Chen** [2], **Saru Kumari** [3] **and Chienming Chen** [1,*]

1 College of Computer Science and Engineering, Shandong University of Science and Technology, Qingdao 266590, China; wutsuyang@sdust.edu.cn (T.W.); xinglan2021@sdust.edu.cn (X.G.)
2 Department of Computer Science, University of California, Davis, CA 001313, USA; ycch@ucdavis.edu
3 Department of Mathematics, Chaudhary Charan Singh University, Meerut 250004, Uttar Pradesh, India; aru@ccsuniversity.ac.in
* Correspondence: chienmingchen@ieee.org

**Abstract:** At present, the great progress made by the Internet of Things (*IoT*) has led to the emergence of the Internet of Drones (*IoD*). *IoD* is an extension of the *IoT*, which is used to control and manipulate drones entering the flight area. Now, the fifth-generation mobile communication technology (5*G*) has been introduced into the *IoD*; it can transmit ultra-high-definition data, make the drones respond to ground commands faster and provide more secure data transmission in the *IoD*. However, because the drones communicate on the public channel, they are vulnerable to security attacks; furthermore, drones can be easily captured by attackers. Therefore, to solve the security problem of the *IoD*, Hussain et al. recently proposed a three-party authentication protocol in an *IoD* environment. The protocol is applied to the supervision of smart cities and collects real-time data about the smart city through drones. However, we find that the protocol is vulnerable to drone capture attacks, privileged insider attacks and session key disclosure attacks. Based on the security of the above protocol, we designed an improved protocol. Through informal analysis, we proved that the protocol could resist known security attacks. In addition, we used the real-oracle random model and ProVerif tool to prove the security and effectiveness of the protocol. Finally, through comparison, we conclude that the protocol is secure compared with recent protocols.

**Keywords:** authentication; Internet of Drones; 5G networks; cryptanalysis; lightweight

## 1. Introduction

In the past decade, the development of artificial intelligence [1–3] and network has witnessed significant advances. A network called the Internet of Things (*IoT*) has emerged, which connects physical objects to the network and realizes a comprehensive information interaction between objects and between people and objects [4–8]. The *IoT* deploys sensors in a specific network area to collect real-time information to meet various user needs. In the *IoT*, the data transmission mode is WiFi or Bluetooth, but the communication distance of these two transmission modes is limited. Therefore, researchers have proposed using the base station for data transmission. The fourth-generation mobile communication technology (4*G*) is suitable for scenarios involving a large amount of data, strong mobility and remote use areas. However, in some cases, data transmission is unstable and the speed is not too fast, which makes meeting user needs difficult. Now, a new mobile communication network, the fifth-generation mobile communication technology (5*G*) has appeared [9]. The application of 5*G* technology to the *IoT* environment increases the capacity of access equipment, expands the coverage area of the signal and improves the stability of signal [10,11].

Recently, drones, which are aircraft managed by a control station, have been introduced [12,13]. The physical structure of drones includes the sensor, receiver, recorder, communication module and actuator. These devices have always been used in the military field and are now widely used in civil fields, including aerial photography, express

transportation, disaster relief and power patrol inspection. These applications have been used to make people's daily life more efficient. Due to their mobile characteristics, drones have been introduced into the *IoT* and form a special network called the Internet of Drones (*IoD*) [14–17]. The development of communication technology in the *IoD* is similar to that in the *IoT*. Due to the limited communication distance of WiFi or Bluetooth, the flight range of drones is significantly limited. Consequently, researchers have proposed a new communication mode, namely, network-connected drones, which uses base stations to connect and control drones. However, when 4*G* is used in some specific scenarios (face recognition, high-altitude requirements, etc.), its resolution is not very clear and positioning may be inaccurate, which hinders meeting user needs. Now, some studies in recent times have combined 5*G* with *IoD* [18,19]. The high-broadband characteristics of 5*G* can transmit ultra-high-definition data and its low delay can make responding to ground commands faster, operate more accurately and provide more secure data transmission. The universal *IoD* architecture is shown in Figure 1. This architecture has been mentioned in many works of literature [20,21]. The *IoD* architecture consists of four entities: user, server, drone and control room. The user obtains real-time information captured by the drones in the flight area and the control room formulates the flight mission of the drones and controls the flight area and flight altitude. Each drone is deployed in different flight areas, using built-in sensors to detect the physical phenomena of the target, or built-in cameras to capture the target video. When users want to obtain the information captured by drones in a certain area, they send a data request to the servers. The servers find the drones in the corresponding area and ask to upload the captured information, then transmit the data through 5*G* technology to meet user needs.

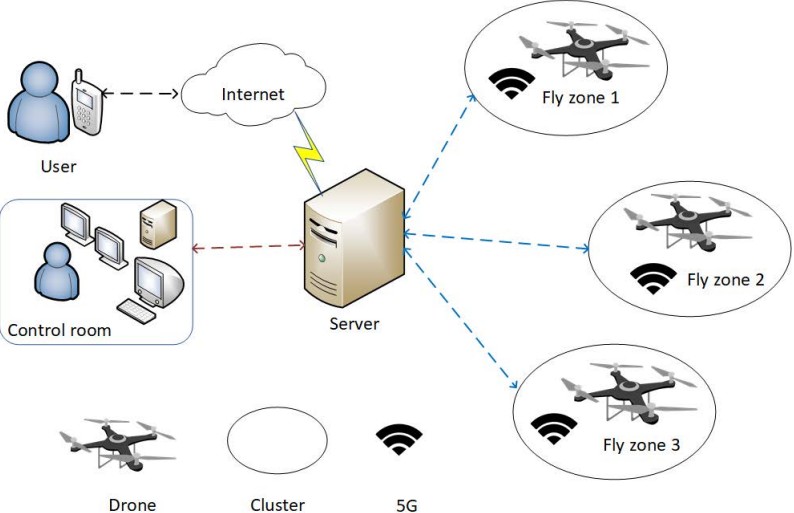

**Figure 1.** *IoD* architecture.

However, in the *IoD* environment, because drones communicate on the public channel, they may be attacked by attackers, such as replay attacks [22], impersonation attacks [20,21], or man-in-the-middle attacks [21,23]. Moreover, the real-time sensitive data in transmission need to be kept confidential [24,25]. Drones flying in a certain area are also susceptible to being captured by attackers and the secret values stored in the memory would also be exposed [23,26]. Therefore, these problems make it necessary to design security protocols to ensure normal communication. Due to the relatively weak computing and storage capabilities of drones, the designed protocol should also meet the lightweight requirements. Recently, Hussain et al. [21] proposed a three-party authentication protocol in an *IoD* environment. Because the computing power of drones is relatively weak, the protocol is lightweight and conforms to the scenario of using the drone. The protocol is applied to the supervision of the smart city, users who want to obtain real-time data about smart cities collected by drones can only transmit data after both parties have been authenticated.

However, we find that Hussain et al.'s protocol is vulnerable to drone capture attacks and privileged insider attacks. The attacker can also impersonate the user by obtaining the information in the server memory to complete a session. Based on the security problems of Hussain et al.'s protocol, it is necessary to design an enhanced protocol to ensure communication security. To solve these security problems, we improved Hussain et al.'s protocol by proposing an enhanced authentication protocol. Based on the relatively weak computing power of drones, the protocol uses only lightweight primitives. Through informal analysis, we proved that the protocol can resist known security attacks. In addition, we used the real-oracle random (ROR) model and ProVerif tool to prove the security and effectiveness of the protocol. Finally, by comparing the proposed protocol with other available protocols, we show that the effectiveness of the proposed protocol.

The rest of this paper is structured as follows: In Section 2, we review the recent research results on the drone communication authentication protocol. In Section 3, we briefly describe the protocol of Hussain et al. [21] and we point out their security problems. We show the specifics of the proposed protocol in Section 4. In Section 5, we use the ROR model, ProVerif tool and informal security analysis to prove the security and effectiveness of the proposed protocol. In Section 6, we compare the proposed protocol with recent protocols and conclude that our protocol is better in terms of security. Finally, we present our conclusions in Section 7.

## 2. Related Work

A variety of studies have focused on designing authentication and key agreement (AKA) protocols for *IoT*. In 2014, Turkanovic et al. [27] designed an authentication protocol to ensure secure communication in the *IoT* environment. The protocol has a low computational cost and only uses lightweight primitives. However, in 2016, Farash et al. [28] pointed out that Turkanovic et al.'s protocol [27] could not provide anonymity of users and sensors and could not resist session key disclosure attacks, stolen smart card attacks and sensor node impersonation attacks. This showed that Turkanovic et al.'s protocol [27] was unsafe, which was contrary to their statement of security at that time. At the same time, Farash et al. [28] proposed an enhanced protocol based on Turkanovic et al.'s [27] and claimed that the protocol was secure. However, Amin et al. [29] showed that the protocol of Farash et al. [28] was vulnerable to off-line password guessing attacks, user impersonation attacks and temporary information disclosure attacks. Amin et al. [29] proposed a three-factor authentication protocol, which could realize anonymous protection. Later, research on the combination of *IoT* and 5G was mentioned in the literature [30–32]. In 2019, Lee et al. [30] designed a cross-layer protocol based on the physical layer and cryptography authentication. In the same year, Jangirala et al. [31] proposed an authentication protocol based on blockchain. The protocol uses radio frequency identification (RFID) technology and bit rotation operation to realize the security authentication of the *IoT* environment. In 2020, Minahil et al. [32] designed an authentication protocol for *IoT* applications. The protocol uses the hash function and elliptic curve encryption (ECC) point addition operation to realize mutual authentication between users and servers.

Recently, research on the *IoD* has been widely conducted [33–35]. All the architectures proposed in the above literature are for users to communicate with drones, but other architectures of drones may need to be discussed in real life, such as the combination and communication between drones and external smart devices in the *IoT* environment [36,37]. Then, researchers designed an AKA protocol for *IoD*. Bera et al. [38] designed a blockchain-based control protocol that not only uses the hash function but also uses expensive operation primitives, such as ECC and digital signature. Tian et al. [39] designed an AKA based on privacy protection that uses expensive digital signature and modular multiplication. In addition, Li et al. [40] proposed a secure authentication mechanism based on ECC and claimed to be lightweight, but the mechanism encrypts messages through a public key infrastructure (PKI) mechanism, which is cost-intensive. Ever et al. [20] designed a security protocol, which realizes the mutual authentication between users and drones. The protocol [20] uses bilinear pairing and ECC, which has a large computational cost.

Moreover, the protocol [20] cannot provide user anonymity and untraceability and cannot resist drone capture attacks. Hussain et al. [21] designed a protocol for smart cities using drones. The protocol [21] uses symmetric encryption operation and cannot resist privileged insider attacks, impersonation attacks and drone capture attacks. The protocols mentioned above have high costs and make it difficult to meet the needs of lightweight computing primitives for drones due to their weak computing power.

Subsequently, researchers began to design AKA protocols for the *IoD* from a lightweight perspective. In 2019, Srinivas et al. [23] designed an AKA based on temporal credentials and claimed that the protocol [23] could resist known attacks. However, Ali et al. [41] found that Srinivas et al.'s protocol [23] could not provide user anonymity and could not resist mutual authentication, stolen verification attacks. In the same year, Wizard et al. [20] designed a protocol based on anonymity, but the protocol [20] could not achieve mutual authentication and was vulnerable to privileged insider attacks and impersonation attacks. In 2020, Chen et al. [22] proposed a privacy protection authentication protocol for drone communication, but the protocol [22] could not resist temporary information disclosure and replay attacks. In 2020, Zhang et al. [42] proposed a key agreement protocol, which is lightweight and suitable for the *IoD* environment. However, the protocol [42] could not resist the stolen smart card attacks nor provide untraceability.

In recent years, 5*G* technology has been introduced into the *IoD* to improve the clarity and security of data transmission. In 2020, Abdel et al. [43] proposed an authentication inspired by the second factor, which triggers the second factor for authentication. In 2021, Abdel et al. [18] designed a signature-based authentication in a 5*G* network. In the same year, Alladi et al. [19] designed an AKA protocol, also known as Drone-MAP. The authentication protocol [19] was based on a 5*G* network and uses a physical unclonable function (PUF) to realize mutual authentication between the drone and 5*G* base station. Some important related works are summarized in Table 1.

**Table 1.** The summary of authentication protocols.

| Protocols | Cryptographic Techniques | Limitations |
| --- | --- | --- |
| Turkanovic et al. [27] | (1) Utilizes one-way hash function<br>(2) Based on smart card | (1) Does not resist session key disclosure attacks<br>(2) Does not provide user anonymity<br>(3) Does not resist sensor node impersonation attacks |
| Farash et al. [28] | (1) Utilizes one-way hash function<br>(2) Based on smart card<br>(3) Two-factor | (1) Does not resist off-line password guessing attacks<br>(2) Does not resist user impersonation attacks<br>(3) Does not resist temporary information disclosure attacks |
| Zhang et al. [42] | (1) Utilizes one-way hash function<br>(2) Based on smart card<br>(3) Two-factor | (1) Does not resist stolen smart card attacks<br>(2) Does not provide untraceability |
| Chen et al. [22] | (1) Utilizes one-way hash function<br>(2) Utilizes ECC<br>(3) Utilizes asymmetric encryption | (1) Does not resist temporary information disclosure attacks<br>(2) Does not resist replay attacks |
| Srinivas et al. [23] | (1) Utilizes one-way hash function<br>(2) Three-factor | (1) Does not resist privileged insider attacks<br>(2) Does not resist drone capture attacks<br>(3) Does not provide user anonymity and untraceability |
| Wazid et al. [20] | (1) Utilizes one-way hash function<br>(2) Three-factor | (1) Does not resist privileged insider attacks<br>(2) Does not resist impersonation attacks<br>(3) Does not provide mutual authentication |
| Ever et al. [26] | (1) Utilizes one-way hash function<br>(2) Utilizes bilinear pairing<br>(3) Utilizes ECC | (1) Does not resist privileged insider attacks<br>(2) Does not resist drone capture attacks<br>(3) Does not provide user anonymity and untraceability |
| Hussain et al. [21] | (1) Utilizes one-way hash function<br>(2) Based on symmetric encryption<br>(3) Three-factor | (1) Does not resist privileged insider attacks<br>(2) Does not resist impersonation attacks<br>(3) Does not resist drone capture attacks |

### 3. Security Analysis of Hussain et al.'s Protocol

*3.1. Review of Hussain et al.'s Protocol*

In this section, we briefly review the protocol of Hussain et al. [21]. The protocol consists of three entities: user ($U_i$), drone ($D_j$) and server ($S$). The protocol has three phases: predeployment phase, user registration phase and login authentication phase. The symbols used in this protocol are shown in Table 2.

**Table 2.** Notations used in the protocol.

| Symbol | Description |
| --- | --- |
| $U_i$ | The $i$-th user |
| $D_j$ | The $j$-th drone |
| $S$ | Server |
| $ID_i, ID_j, ID_S$ | Identities of $U_i$, $D_j$ and $S$ |
| $TID_i$ | Temporary identities of $U_i$ |
| $PSW_i$ | Password of $U_i$ |
| $K_S$ | Secret key of $S$ |
| $SK$ | Session key |

#### 3.1.1. Predeployment Phase

$S$ selects an identity $ID_j$ for all drones $D_j$ before deployment, computes the value $N_j = h(ID_j \| K_s)$, then saves $\{ID_j, N_j\}$ to the memory of drones $D_j$ and, finally, saves the $ID_j$ to its own memory.

#### 3.1.2. User Registration Phase

(1) First, user $U_i$ selects its identity $ID_i$ and then sends the identity $ID_i$ to $S$ through the secure channel.

(2) After receiving $ID_i$, $S$ selects the random number $n$, computes $RID_i = E_{K_s}(ID_i \| n)$ and $N_i = h(ID_i \| n \| K_s)$ and then saves the identity $ID_i$ to its database. Finally, $S$ generates the random number $M$ and sends message $\{RID_i, N_i, ID_j, M\}$ to $U_i$.

(3) After receiving the message $\{RID_i, N_i, ID_j, M\}$ sent by $S$, $U_i$ selects the password $PSW_i$, biometric $BIO_i$ and the random number $r_i$, then computes $Gen(BIO_i) = (\sigma_i, \tau_i)$, $RID_i' = RID_i \oplus (PSW_i \| \sigma_i)$, $ID_j' = ID_j \oplus (ID_i \| PSW_i \| \sigma_i)$, $N_i' = N_i \oplus (ID_i \| \sigma_i)$, $RPW_i = (PSW_i \| r_i)$, $M' = M \oplus (ID_i \| PSW_i \| \sigma_i)$, $R_i = r_i \oplus (PSW_i \| ID_i \| \sigma_i)$ and $P_i = (M \| RID_i \| RPW_i \| \sigma_i)$. Finally, $U_i$ stores $\{RID_i', ID_j', N_i', M', R_i, P_i, \tau_i, Gen(), Rep(), h()\}$ in mobile device $MD_i$.

#### 3.1.3. Login and Authentication Phase

(1) First, $U_i$ enters the identity $ID_i$, password $PSW_i'$ and biometric $BIO_i'$ into $MD_i$ and $MD_i$ computes $\sigma_i' = Rep(BIO_i', \tau_i)$, $RID_i = RID_i' \oplus (PSW_i' \| \sigma_i')$, $ID_j = ID_j' \oplus (ID_i \| PSW_i' \| \sigma_i')$, $N_i = N_i' \oplus (ID_i \| \sigma_i')$, $r_i = R_i \oplus (PSW_i' \| ID_i \| \sigma_i')$, $RPW_i' = (PSW_i' \| r_i)$, $M = M' \oplus (ID_i \| PSW_i' \| \sigma_i')$, $P_i = (M \| RID_i \| RPW_i' \| \sigma_i')$. Then, $MD_i$ compares $P_i' \overset{?}{=} P_i$. If equal, it means that $U_i$ successfully logs in to $MD_i$. Otherwise, the login fails. After a successful login, $MD_i$ selects a random number $r_1$ and timestamp $T_1$, then computes $A_1 = RID_i$, $A_2 = ID_j \oplus h(N_i \| ID_i \| T_1)$, $A_3 = h(ID_s \| N_i \| T_1) \oplus r_1$ and $A_4 = h(ID_i \| ID_s \| ID_j \| N_i \| r_1 \| T_1)$. Finally, $U_i$ sends the authentication request $M_1 = \{A_1, A_2, A_3, A_4, T_1\}$ to $S$.

(2) After receiving the authentication request $M_1$ from $U_i$, $S$ first verifies the freshness of the timestamp $T_1$. If the time has been exceeded, the authentication is terminated. Otherwise, $S$ computes $(ID_i \| n) = D_{Ks}(A_1)$ to verify whether $ID_i$ is registered. If it is registered, $S$ computes $N_i = h(ID_i \| n \| K_s)$, $ID_j = A_2 \oplus h(N_i \| ID_i \| T_1)$, $r_1' = h(ID_s \| N_i \| T_1) \oplus A_3$ and $A_4' = h(ID_i \| ID_s \| ID_j \| N_i \| r_1' \| T_1)$. Then,

$S$ compares $A_4' \stackrel{?}{=} A_4$. If not equal, it means that $U_i$ is illegal. Otherwise, $S$ selects the random numbers $r_2$, $r_i^{new}$ and timestamp $T_2$ and computes $N_j = h(ID_j \| K_s)$, $A_5 = h(N_j \| ID_j) \oplus h(ID_j \| r_1 \| r_2)$, $A_6 = h(N_j \| T_2) \oplus ID_i$, $A_7 = h(N_j \| ID_j \| h(ID_s \| r_1 \| r_2) \| T_2)$ and $A_8 = E_{K_s}(ID_i \| r_i^{new}) \oplus h(N_i \| ID_i \| RID_i)$. Finally, $S$ sends the message $M_2 = \{A_5, A_6, A_7, A_8, T_2\}$ to drone $D_j$.

(3) After receiving the message $M_2$, $D_j$ first verifies the freshness of $T_2$. If the time has not been exceeded, it computes $ID_i = h(N_j \| T_2) \oplus A_6$, $A_9 = h(ID_j \| N_j) \oplus A_5$, $A_{10} = h(N_j \| ID_j \| A_9 \| T_2)$. Then, $D_j$ compares $A_{10} \stackrel{?}{=} A_7$. If not equal, it means that $S$ is illegal. Otherwise, $D_j$ selects the random number $r_3$ and timestamp $T_3$ and computes $A_{11} = h(ID_j \| ID_i \| T_3) \oplus r_3$, $SK = h(A_9 \| r_3 \| ID_i \| ID_j)$, $A_{12} = h(ID_i \| ID_j \| r_3) \oplus A_9$ and $A_{13} = h(SK \| T_3)$. Finally, $D_j$ sends the message $M_3 = \{A_{11}, A_{12}, A_{13}, A_8, T_3\}$ to $U_i$.

(4) After receiving the message $M_3$, $U_i$ first verifies the freshness of $T_3$. If the time has not been exceeded, it computes $r_3' = h(ID_j \| ID_i \| T_3) \oplus A_{11}$, $A_9' = h(ID_i \| ID_j \| r_3) \oplus A_{12}$, $SK = h(A_9' \| r_3' \| ID_i \| ID_j)$ and $A_{14} = h(SK \| T_3)$. Then, $U_i$ compares $A_{14} \stackrel{?}{=} A_{13}$. If not equal, it means that $D_j$ is illegal and the authentication is terminated. Otherwise, the authentication is successful. Finally, $U_i$ updates $\overline{RID_i} = A_8 \oplus h(N_i \| ID_i \| RID_i)$ and $RID_i = \overline{RID_i}$.

### 3.2. Cryptanalysis of Hussain et al.'s Protocol

In this part, we point out that the protocol of Hussain et al. [21] is vulnerable to drone capture attacks, session key disclosure attacks and drone impersonation attacks.

### 3.2.1. Adversary Model

We briefly describe the capabilities of the adversary ($A$) and use the $D$–$Y$ model according to the literature [44–46]. The capabilities are described in detail as follows:

(1) $A$ can intercept, modify and eavesdrop messages transmitted on public channels.
(2) $A$ can obtain information stored in the server.
(3) $A$ can extract the private value in the memory of the captured drones.

### 3.2.2. Drone Capture Attacks

We assume that $A$ can capture drones $D_j$ and obtain the value $\{ID_j, N_j\}$ stored in the memory of $D_j$. $A$ can compute the session key $SK$ through the following steps:

(1) $A$ first intercepts $\{A_5, A_6, T_2\}$ in $M_2$ and $\{A_{11}, T_3\}$ in $M_3$ transmitted by the common channel.
(2) $A$ can compute $ID_i$, $A_9$ and $r_3$ through $ID_i = h(N_j \| T_2) \oplus A_6$, $A_9 = h(ID_j \| N_j) \oplus A_5$ and $r_3 = h(ID_j \| ID_i \| T_3) \oplus A_{11}$.
(3) $A$ can successfully compute $SK = h(A_9 \| r_3 \| ID_i \| ID_j)$.

Therefore, the protocol of Hussain et al. [21] cannot resist drone capture attacks.

### 3.2.3. Privileged Insider Attacks

We assume that $A$ obtains $K_s$ stored in the server. Based on this attack, there are two security vulnerabilities.

A. Session Key Disclosure Attacks

(1) $A$ first intercepts $\{A_2, A_5, A_{11}, T_1, T_2\}$ transmitted by the common channel.
(2) $A$ can obtain $ID_i$ and $n$ by computing $(ID_i \| n) = D_{Ks}(A_1)$ with the value of $K_s$. Then, $A$ can compute $N_j = h(ID_j \| K_s)$, $ID_j = A_2 \oplus h(N_i \| ID_i \| T_1)$, $A_9 = h(ID_j \| N_j) \oplus A_5$ and $r_3 = h(ID_j \| ID_i \| T_3) \oplus A_{11}$.
(3) $A$ can successfully compute $SK = h(A_9 \| r_3 \| ID_i \| ID_j)$.

Therefore, the protocol of Hussain et al. [21] cannot resist session key disclosure attacks.

B. Drone Impersonation Attacks

Similar to the session key disclosure attacks mentioned above, this attack is also based on privileged insider attacks. $A$ can obtain $\{ID_i, ID_j, N_j, A_9, A_8\}$.

(1) After receiving the message $M_2$ sent by $S$, $A$ selects the random number $r_3^*$ and timestamp $T_3^*$ and computes $A_{11}^* = h(ID_j \parallel ID_i \parallel T_3^*) \oplus r_3^*$, $SK = h(A_9 \parallel r_3^* \parallel ID_i \parallel ID_j)$, $A_{12}^* = h(ID_i \parallel ID_j \parallel r_3^*) \oplus A_9$ and $A_{13}^* = h(SK \parallel T_3^*)$. Finally, $D_j$ sends the message $M_3^* = \{A_{11}^*, A_{12}^*, A_{13}^*, A_8, T_3^*\}$ to $U_i$.

(2) After receiving the message $M_3^*$, $U_i$ first verifies the freshness of $T_3^*$ and computes $r_3^* = h(ID_j \parallel ID_i \parallel T_3^*) \oplus A_{11}^*$, $A_{12}^* = h(ID_i \parallel ID_j \parallel r_3^*) \oplus A_{12}^*$, $SK = h(A_9 \parallel r_3^* \parallel ID_i \parallel ID_j)$ and $A_{14}^* = h(SK \parallel T_3^*)$. $U_i$ compares $A_{14}^* \overset{?}{=} A_{13}^*$. Then, $U_i$ successfully authenticates $A$ and establishes session key $SK$. Finally, $U_i$ updates $\overline{RID_i} = A_8 \oplus h(N_i \parallel ID_i \parallel RID_i)$ and $RID_i = \overline{RID_i}$.

Therefore, $A$ can impersonate a legitimate $D_j$ to complete authentication with $U_i$; the protocol of Hussain et al. [21] cannot resist drone impersonation attacks.

## 4. The Proposed Protocol

To improve the security of Hussain et al.'s protocol [21], we propose an improved protocol based on the architecture shown in Figure 1. The protocol consists of three entities: $U_i$, $D_j$ and $S$. The protocol has three phases: drone registration phase, user registration phase and login authentication phase.

### 4.1. Drone Registration Phase

$D_j$ registers with $S$. The registration phase is shown in Figure 2. The registration steps are as follows.

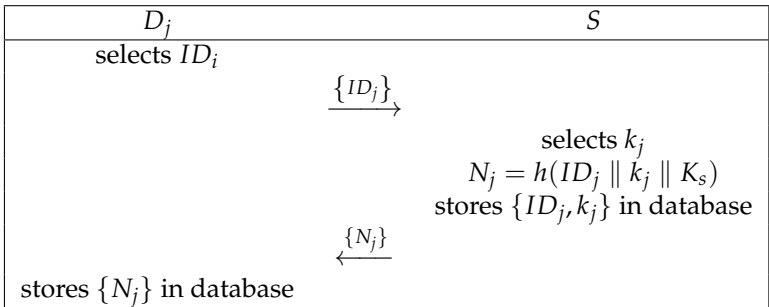

**Figure 2.** $D_j$ registration phase.

(1) First, drone $D_j$ selects its identity $ID_j$ and then sends the identity $ID_j$ to $S$ through the secure channel.

(2) After receiving $ID_j$, $S$ selects the random number $k_j$, computes $N_j = h(ID_j \parallel k_j \parallel K_s)$ and then saves $\{ID_j, k_j\}$ to its database. Finally, $S$ sends message $\{N_j\}$ to $D_j$.

(3) After receiving the message $\{N_j\}$ sent by $S$, $D_j$ stores $\{N_j\}$ in its database.

### 4.2. User Registration Phase

$U_i$ registers with $S$. The registration phase is shown in Figure 3. The registration steps are as follows.

| $U_i$ | $S$ |
|---|---|

**Figure 3.** $U_i$ registration phase.

(1) First, user $U_i$ selects $ID_i, PSW_i, BIO_i, r$ and computes $Gen(BIO_i) = (\sigma_i, \tau_i)$. Then, $U_i$ sends identity $ID_i$ to $S$ through the secure channel.

(2) After receiving $ID_i$, $S$ selects the random number $k_i$ and $TID_i$, computes $RPW_i = (PSW_i \| r)$, $RID_i = h(ID_i \| k_i)$, $N_i = h(RID_i \| k_i \| K_s)$ and $RID_i^* = RID_i \oplus h(k_i \| K_s)$ and then saves the identity $\{TID_i, RID_i^*, k_j\}$ to its database. Finally, $S$ sends message $\{RID_i, N_j, ID_j, TID_i\}$ to $U_i$.

(3) After receiving the message $\{RID_i, N_i, ID_j, TID_i\}$ sent by $S$, $U_i$ computes $RID_i' = RID_i \oplus (ID_i \| \sigma_i)$, $N_i' = N_i \oplus (PSW_i \| \sigma_i)$, $ID_j' = ID_j \oplus (ID_i \| PSW_i \| r)$ and $P_i = (ID_i \| RPW_i \| \sigma_i)$. Finally, $U_i$ stores $\{RID_i', ID_j', N_i', r, P_i, TID_i, \tau_i, Gen(), Rep(), h()\}$ in mobile device $MD_i$.

*4.3. Login and Authentication Phase*

In the login and authentication phase, $U_i$ and $D_j$ achieve mutual authentication and establish session key $SK$ with the help of $S$. The login and authentication phase is shown in Figure 4 and the steps are as follows:

(1) First, $U_i$ enters identity $ID_i$, password $PSW_i$ and biometric $BIO_i$ into $MD_i$ and $MD_i$ computes $\sigma_i' = Rep(BIO_i, \tau_i)$, $RPW_i = (PSW_i \| r)$ and $P_i^* = (ID_i \| RPW_i \| \sigma_i')$. Then, $MD_i$ compares $P_i' \stackrel{?}{=} P_i$. If they are equal, it means that $U_i$ successfully logs in to $MD_i$. Otherwise, the login fails. After a successful login, $MD_i$ computes $RID_i = RID_i' \oplus (PSW_i \| \sigma_i')$, $ID_j = ID_j' \oplus (ID_i \| PSW_i \| r)$ and $N_i = N_i' \oplus (PSW_i \| \sigma_i')$. Then, $MD_i$ selects the random number $r_1$ and timestamp $T_1$ and computes $A_1 = h(RID_i \| T_1) \oplus r_1$, $A_2 = ID_j \oplus h(N_i \| RID_i \| T_1)$ and $V_1 = h(RID_i \| ID_j \| r_1 \| T_1)$. Finally, $U_i$ sends the authentication request $M_1 = \{A_1, A_2, V_1, TID_i, T_1\}$ to $S$.

(2) After receiving the authentication request $M_1$ from $U_i$, $S$ first verifies the freshness of the timestamp $T_1$. If the time has been exceeded, the authentication is terminated. Otherwise, $S$ searches $\{RID_i^*, k_j\}$ according to $TID_i$ and computes $RID_i = RID_i^* \oplus h(k_i \| K_s)$, $N_i = h(RID_i \| k_i \| K_s)$, $r_1 = h(RID_i \| T_1) \oplus A_1$, $ID_j = A_2 \oplus h(N_i \| RID_i \| T_1)$ and $V_1' = h(RID_i \| ID_j \| r_i \| T_1)$. Subsequently, $S$ compares $V_1' \stackrel{?}{=} V_1$. If they are not equal, it means that $U_i$ is illegal. Otherwise, $S$ selects the random number $r_2$ and timestamp $T_2$ and computes $N_j = h(ID_j \| k_j \| K_s)$, $A_3 = r_2 \oplus h(ID_j \| N_j)$, $A_4 = RID_i \oplus h(ID_j \| N_j \| T_2)$ and $V_2 = h(ID_j \| N_j \| r_2 \| T_2)$. Finally, $S$ sends the message $M_2 = \{A_1, A_3, A_4, V_2, T_1, T_2\}$ to drone $D_j$.

(3) After receiving the message $M_2$, $D_j$ first verifies the freshness of $T_2$. If the time has not been exceeded, it computes $RID_i = h(ID_j \| N_j \| T_2) \oplus A_4$, $r_2 = h(ID_j \| N_j) \oplus A_3$

and $V_2' = h(ID_j \parallel N_j \parallel r_2 \parallel T_2)$. Then, $D_j$ compares $V_2' \overset{?}{=} V_2$. If they are not equal, it means that $S$ is illegal. Otherwise, $D_j$ selects the random number $r_3$ and timestamp $T_3$ and computes $r_1 = h(RID_i \parallel T_1) \oplus A_1$, $SK = h(RID_i \parallel ID_j \parallel r_1 \parallel r_2 \parallel r_3)$, $A_5 = h(RID_i \parallel T_3) \oplus r_3$, $A_6 = h(ID_j \parallel T_3) \oplus r_2$ and $V_3 = h(SK \parallel ID_j \parallel RID_i \parallel r_1 \parallel r_3)$. Finally, $D_j$ sends the message $M_3 = \{A_5, A_6, V_3, T_3\}$ to $U_i$.

(4) After receiving the message $M_3$, $U_i$ first verifies the freshness of $T_3$. If the time has not been exceeded, it computes $r_3 = h(RID_i \parallel T_3) \oplus A_5$, $r_2 = h(ID_j \parallel T_3) \oplus A_6$, $SK = h(RID_i \parallel ID_j \parallel r_1 \parallel r_2 \parallel r_3)$ and $V_3' = h(SK \parallel ID_j \parallel RID_i \parallel r_1 \parallel r_3)$. Then, $U_i$ compares $V_3' \overset{?}{=} V_3$. If they are not equal, it means that $D_j$ is illegal and the authentication is terminated. Otherwise, the authentication is successful.

| $U_i$ | $S$ | $D_j$ |
|---|---|---|
| Inputs $ID_i\ PSW_i$, imprints $BIO_i$ | | |
| $\sigma_i' = Rep(BIO_i, \tau_i)$ | | |
| $RPW_i = (PSW_i \parallel r)$ | | |
| $P_i^* = (ID_i \parallel RPW_i \parallel \sigma_i')$ | | |
| $Checks P_i' \overset{?}{=} P_i$ | | |
| $RID_i = RID_i' \oplus (PSW_i \parallel \sigma_i')$ | | |
| $ID_j = ID_j' \oplus (ID_i \parallel PSW_i \parallel r)$ | | |
| $N_i = N_i' \oplus (PSW_i \parallel \sigma_i')$ | | |
| selects $r_1$ and $T_1$ | | |
| $A_1 = h(RID_i \parallel T_1) \oplus r_1$ | | |
| $A_2 = ID_j \oplus h(N_i \parallel RID_i \parallel T_1)$ | | |
| $V_1 = h(RID_i \parallel ID_j \parallel r_1 \parallel T_1)$ | | |
| $\xrightarrow{M_1 = \{A_1, A_2, V_1, TID_i, T_1\}}$ | | |
| | Checks $\lvert T_1 - T_c \rvert \leqq \Delta T$ | |
| | searches $\{RID_i^*, k_j\}$ according to $TID_i$ | |
| | $RID_i = RID_i^* \oplus h(k_i \parallel K_s)$ | |
| | $N_i = h(RID_i \parallel k_i \parallel K_s)$ | |
| | $r_1 = h(RID_i \parallel T_1) \oplus A_1$ | |
| | $ID_j = A_2 \oplus h(N_i \parallel RID_i \parallel T_1)$ | |
| | $V_1' = h(RID_i \parallel ID_j \parallel r_i \parallel T_1)$ | |
| | Checks $V_1' \overset{?}{=} V_1$. Selects $r_2, T_2$ | |
| | $N_j = h(ID_j \parallel k_j \parallel K_s)$ | |
| | $A_3 = r_2 \oplus h(ID_j \parallel N_j)$ | |
| | $A_4 = RID_i \oplus h(ID_j \parallel N_j \parallel T_2)$ | |
| | $V_2 = h(ID_j \parallel N_j \parallel r_2 \parallel T_2)$ | |
| | $\xrightarrow{M_2 = \{A_1, A_3, A_4, V_2, T_1, T_2\}}$ | |
| | | Checks $\lvert T_2 - T_c \rvert \leqq \Delta T$ |
| | | $RID_i = h(ID_j \parallel N_j \parallel T_2) \oplus A_4$ |
| | | $r_2 = h(ID_j \parallel N_j) \oplus A_3$ |
| | | $V_2' = h(ID_j \parallel N_j \parallel r_2 \parallel T_2)$ |
| | | Checks $V_2' \overset{?}{=} V_2$ |
| | | $r_1 = h(RID_i \parallel T_1) \oplus A_1$ |
| | | selects $r_3$ and $T_3$ |
| | | $SK = h(RID_i \parallel ID_j \parallel r_1 \parallel r_2 \parallel r_3)$ |
| | | $A_5 = h(RID_i \parallel T_3) \oplus r_3$ |
| | | $A_6 = h(ID_j \parallel T_3) \oplus r_2$ |
| | | $V_3 = h(SK \parallel ID_j \parallel RID_i \parallel r_1 \parallel r_3)$ |
| | | $\xleftarrow{M_3 = \{A_5, A_6, V_3, T_3\}}$ |
| Checks $\lvert T_3 - T_c \rvert \leqq \Delta T$ | | |
| $r_3 = h(RID_i \parallel T_3) \oplus A_5$ | | |
| $r_2 = h(ID_j \parallel T_3) \oplus A_6$ | | |
| $SK = h(RID_i \parallel ID_j \parallel r_1 \parallel r_2 \parallel r_3)$ | | |
| $V_3' = h(SK \parallel ID_j \parallel RID_i \parallel r_1 \parallel r_3)$ | | |
| checks $V_3' \overset{?}{=} V_3$ | | |

**Figure 4.** Login and authentication phase.

## 5. Security Analysis

### 5.1. Formal Security Analysis

In this part, we show how we used the ROR model to analyze the security of the proposed protocol. The ROR model was proposed by Canetti et al. [47,48]. The ROR model

was used to judge the security of the protocol by obtaining the probability of successfully cracking the session key *SK* through different game rounds.

### 5.1.1. ROR Model

The protocol consists of three entities: $U_i$, $D_j$ and $S$. In the ROR model, $\Pi_{U_i}^x$, $\Pi_{D_j}^y$ and $\Pi_S^z$ represent the x-th instance of $U_i$, the y-th instance of $D_j$ and the z-th instance of $S$, respectively. Let us suppose that $A$ has the following query capabilities: $Q = \{\Pi_{U_i}^x, \Pi_{D_j}^y, \Pi_S^z\}$.

(1) $Execute(Q)$: By executing this query, $A$ can intercept messages transmitted among $U_i$, $D_j$ and $S$ on the common channel.

(2) $Send(Q, M)$: By executing this query, $A$ can send message $M$ to $Q$ and receive a response from $Q$.

(3) $Hash(string)$: Through executing this query, $A$ can enter a string and return its hash value.

(4) $Corrupt(Q)$: By executing this query, $A$ can obtain a party's private value, such as long-term key, parameters stored in a smart card, or temporary information.

(5) $Test(Q)$: By executing this query, $A$ flips a coin $C$. If $C = 1$, $A$ can obtain the correct *SK*. If $C = 0$, $A$ can obtain any string of the same length as *SK*.

### 5.1.2. ROR Proof

**Theorem 1.** *In the ROR model, assuming that $A$ can execute the above five queries, the probability that $A$ can break the proposed protocol P in polynomial time is $adv_{\mathcal{A}}^{\mathcal{P}}(\xi) \leq q_{send}/2^{l-2} + 3q_{hash}^2/2^{l-1} + 2max\{C' \cdot q_{send}^{s'}, q_{send}/2^l\}$, where $q_{send}$ refers to the number of queries executed; $q_{hash}$ refers to the number of times hash queries executed; l refers to the bit length of biological information; and C' and s' refer to two constants.*

**Proof.** Our proof consists of seven game rounds, from $GM_0$ to $GM_6$. $Succ_A^{GM_i}(\xi)$ represents the probability that $A$ can win in seven rounds of the game.

$GM_0$: $GM_0$ is the first round of the game and does not start any query operation. This round of the game begins by flipping a coin $C$. Therefore, we can obtain the probability that $A$ can successfully break $P$ as

$$Adv_A^P = |2Pr[Succ_A^{GM_0}] - 1|. \tag{1}$$

$GM_1$: $GM_1$ has one more $Execute(Q)$ operation than $GM_0$ and $A$ intercepts only the message $\{M_1, M_2, M_3\}$ transmitted on the common channel in $GM_1$. Since the values of $ID_j$, $r_1$, $r_2$, $r_3$ and $RID_i$ are unknown, $A$ cannot compute the *SK* through the *test(Q)* query. Therefore, the probability of $GM_1$ is equal to that of $GM_0$.

$$Pr[Succ_A^{GM_1}] = Pr[Succ_A^{GM_0}]. \tag{2}$$

$GM_2$: $GM_2$ has one more $Send(Q)$ operation than $GM_1$. According to Zipf's law [49], we can obtain the probability of $GM_2$ as

$$|Pr[Succ_A^{GM_2}(\xi)] - Pr[Succ_A^{GM_1}(\xi)]| \leq q_{send}/2^l. \tag{3}$$

$GM_3$: $GM_3$ has one more $Hash(Q)$ operation than $GM_2$. According to the birthday paradox, we can obtain the probability of $GM_3$ as

$$|Pr[Succ_A^{GM_3}(\xi)] - Pr[Succ_A^{GM_2}(\xi)]| \leq q_{hash}^2/2^{l+1}. \tag{4}$$

$GM_4$: In this round, the ROR model analyzes two events to prove the security of the protocol. One is to obtain the long-term key $K_S$ of $S$ to prove that the protocol can provide perfect forward security and the other is to obtain the temporary information of an entity

to prove that the protocol can resist the known session-specific temporary information disclose attacks.

(1) Perfect forward security: $A$ uses $\Pi_S^Z$ to obtain the long-term key $K_S$ of $S$ or uses $\Pi_{U_i}^x$ and $\Pi_{D_j}^y$ to obtain the private value used in the registration phase.

(2) Known session-specific temporary information disclose attacks: $A$ uses $\Pi_{U_i}^x$, $\Pi_{D_j}^y$ and $\Pi_S^z$ to obtain random numbers of three parties.

For the previous event, even if $A$ obtains the long-term key $K_S$ of $S$ or the private value used by both in the registration phase, the values of $\{r_1, r_2, r_3, RID_i, ID_j\}$ cannot be computed and $A$ cannot compute the value of $SK$, where $SK = h(RID_i \parallel ID_j \parallel r_1 \parallel r_2 \parallel r_3)$. For the latter event, even if $A$ can obtain $r_1$, the values of $\{r_2, r_3, RID_i, ID_j\}$ are confidential; thus, $SK$ cannot be computed. Similarly, even if $A$ can obtain $r_2$ or $r_3$, the value of $SK$ cannot be computed. We can obtain the probability of $GM_4$ as follows:

$$|Pr[Succ_{\mathcal{A}}^{GM_4}(\xi)] - Pr[Succ_{\mathcal{A}}^{GM_3}(\xi)]| \leq q_{send}/2^l + q_{hash}^2/2^{l+1}. \tag{5}$$

$GM_5$: In $GM_5$, $A$ uses $Corrupt(Q)$ to query the parameters $\{RID_i', ID_j', N_i', M', R_i, P_i, \tau_i\}$, which proves that the protocol can resist offline password guessing attacks. $U_i$ registers with $S$ using password $PSW_i$ and biometric $BIO_i$. $A$ wants to guess $P_i = (M \parallel RID_i \parallel RPW_i \parallel \sigma_i)$, but $ID_i$ and $RPW_i$ are confidential. The probability of $A$ guessing $l$ bits of biological information is $1/2^l$. According to Zipf's law [49], when $q^{send} \leq 10^6$, the probability that $A$ can guess the password is greater than 0.5. Therefore, we can obtain the probability of $GM_5$ as

$$|Pr[Succ_{\mathcal{A}}^{GM_5}(\xi)] - Pr[Succ_{\mathcal{A}}^{GM_4}(\xi)]| \leq max\{C' \cdot q_{send}^{s'}, q_{send}/2^l\} \tag{6}$$

$GM_6$: $GM_6$ is to verify whether protocol $P$ can resist impersonation attacks. $A$ uses $h(RID_i \parallel ID_j \parallel r_1 \parallel r_2 \parallel r_3)$ to query and the game is terminated. Therefore, we can obtain the probability of $GM_6$ as

$$|Pr[Succ_{\mathcal{A}}^{GM_6}(\xi)] - Pr[Succ_{\mathcal{A}}^{GM_5}(\xi)]| \leq q_{hash}^2/2^{l+1}. \tag{7}$$

Because, in $GM_6$, the probability of success and failure is $1/2$, the probability that $A$ can guess $SK$ is

$$Pr[Succ_{\mathcal{A}}^{GM_6}(\xi)] = 1/2. \tag{8}$$

According to the above formula, we can obtain

$$
\begin{aligned}
1/2 Adv_{\mathcal{A}}^{\mathcal{P}}(\xi) &= |Pr[Succ_{\mathcal{A}}^{GM_0}(\xi)] - 1/2| \\
&= |Pr[Succ_{\mathcal{A}}^{GM_0}(\xi)] - Pr[Succ_{\mathcal{A}}^{GM_6}(\xi)]| \\
&= |Pr[Succ_{\mathcal{A}}^{GM_1}(\xi)] - Pr[Succ_{\mathcal{A}}^{GM_6}(\xi)]| \\
&\leq \sum_{i=0}^{5} |Pr[Succ_{\mathcal{A}}^{GM_{i+1}}(\xi)] - Pr[Succ_{\mathcal{A}}^{GM_i}(\xi)]| \\
&= q_{send}/2^{l-1} + 3q_{hash}^2/2^l + max\{C' \cdot q_{send}^{s'}, q_{send}/2^l\}
\end{aligned}
\tag{9}
$$

Therefore, we can obtain

$$Adv_{\mathcal{A}}^{\mathcal{P}}(\xi) \leq q_{send}/2^{l-2} + 3q_{hash}^2/2^{l-1} + 2max\{C' \cdot q_{send}^{s'}, q_{send}/2^l\}. \tag{10}$$

$\square$

### 5.2. ProVerif

We used the formal tool ProVerif to verify the validity of the proposed protocol by modeling, writing code and performing calculations [50,51].

The definition of ProVerif is shown in Figure 5. Here, sch and ch are used to represent the secure channel and common channel, respectively. The parameters and functions of the protocol can be seen from the figure. The functions include h(), mult(), con(), xor(), Gen() and Rep(), which represent hash operations, scalar multiplication, concatenation, XOR, generator and reduction operations, respectively. Figure 6 shows the query operations and events. Here, $SKi$ and $SKj$ represent the session keys of $U_i$ and $D_j$, respectively. "Query attacker" was used to verify whether $A$ could compute $SK$ by intercepting the information on the common channel through query operations. ProVerif contains five events: UserStarted(), UserAuthed(), ServerAcUser(), DroneAcServer() and UserAcDrone().

Figure 7 shows the process of $U_i$, $D_j$ and $S$. $D_j$'s process is similar to $U_i$'s process, so we take the $U_i$'s process as an example. Here, "out(sch,(IDi))" is a registration process initiated by $U_i$ to $S$ in the registration phase and "in(sch,(xRIDi:bitstring,xNi:bitstring,xIDj:bit string,xTIDi:bitstring))" is to simulate $U_i$ to receive the message sent by $S$ in the registration phase. At this time, the registration phase ends. "!()" is $U_i$'s authentication process in the login authentication phase, which means that this phase can occur multiple times, while the registration phase can only occur once. "out(ch,(A1,A2,xTIDi,T1))" means that $U_i$ sends a login request to $S$. "in(ch,(xA5:bitstring,xA6:bitstring,xV3:bitstring,xT3:bitstring))" refers to the $U_i$ who receives the authentication message returned by $S$. As for $S$'s process, it is mainly composed of the "UserReg" $U_i$ registration process, "DroneReg" $D_j$ registration process and "ServerAuth" $S$ authentication process. "UserReg" is the registration process of $S$ in the $U_i$ registration phase, "DroneReg" is the registration process of $S$ in the $D_j$ registration phase, "ServerAuth" is the authentication process adopted by $S$ in the login and authentication phase.

The results of ProVerif are presented in Figure 8. We can see that "Query not attacker (SKi[]) is true", "Query not attacker (SKj[]) is true", "Query inj-event (UserStarted) ==> inj-event (UserAuthed) is true", "Query inj-event(SeverAcUser) ==> inj-event(DroneAcServer) is true" and "Query inj-event(DroneAcServer) ==> inj-event (UserAcDrone) is true". Therefore, it can be concluded that $A$ cannot compute the $SK$ of $U_i$ and drone $D_j$.

```
(*  channel*)
free ch :channel. (*  public channel *)
free sch: channel [private]. (*  secure channel, used for registering *)
(*  shared keys *)
free SKi : bitstring  [private].
free SKj : bitstring  [private].
free SKk : bitstring  [private].
free IDi : bitstring  [private].
(*  constants  *)
free  x:bitstring [private].
(*  functions  &  reductions & equations  *)
fun h(bitstring) :bitstring. (*  hash function *)
fun mult(bitstring,bitstring) :bitstring. (*  scalar multiplication operation *)
fun add(bitstring,bitstring):bitstring. (*  Addition operation *)
fun sub(bitstring,bitstring):bitstring. (*  Subtraction operation *)
fun mod(bitstring,bitstring):bitstring. (*  modulus operation *)
fun con(bitstring,bitstring):bitstring. (*  concatenation operation *)
reduc forall m:bitstring, n:bitstring; getmess(con(m,n))=m.
fun xor(bitstring,bitstring):bitstring. (*  XOR operation *)
equation forall m:bitstring, n:bitstring; xor(xor(m,n),n)=m.
fun Gen(bitstring):bitstring. (*  Generator operation *)
fun Rep(bitstring,bitstring):bitstring.
```

**Figure 5.** The definition in the ProVerif tool.

```
(* queries *)
query attacker(SKi).
query attacker(SKj).
query inj-event(UserStarted()) ==> inj-event(UserAuthed()).
query inj-event(ServerAcUser()) ==> inj-event(DroneAcServer()).
query inj-event(DroneAcServer()) ==> inj-event(UserAcDrone()).

(* event *)
event UserStarted().
event UserAuthed().
event ServerAcUser().
event DroneAcServer().
event UserAcDrone().
```

**Figure 6.** The queries and events in the ProVerif tool.

```
(* ----------------User's process--------------- *)
let ProcessUser=new IDi : bitstring; (* the User's ID *)
new PSWi : bitstring;
new Bioi : bitstring;
new r:bitstring;
let (a: bitstring, b: bitstring)=Gen(Bioi) in
let RPWi=h(con(PSWi,r)) in
out(sch,(IDi));
in(sch,(xRIDi:bitstring,xNi:bitstring,xIDj:bitstring,xTIDi:bitstring));
let RIDi'=xor(xRIDi,h(con(IDi,a))) in
let Ni'=xor(xNi,h(con(PSWi,a))) in
let IDj'=xor(xIDj,h(con(con(IDi,PSWi),r))) in
let Pi=h(con(con(IDi,RPWi),a)) in
!(event UserStarted();
let a=Rep(Bioi,b) in
let RPWi=h(con(PSWi,r)) in
let Pi'=h(con(con(IDi,RPWi),a)) in
if Pi'=Pi   then
let xRIDi=xor(RIDi',h(con(IDi,a))) in
let xNi=xor(Ni',h(con(PSWi,a))) in
let xIDj=xor(IDj',h(con(con(IDi,PSWi),r))) in
new r1:bitstring;
new T1:bitstring;
new RIDi:bitstring;
let A1=xor(r1,h(con(RIDi,T1))) in
let A2=xor(xIDj,h(con(con(xRIDi,xNi),T1))) in
let V1=h(con(con(con(RIDi,xIDj),r1),T1)) in
out(ch,(A1,A2,xTIDi,T1));
in(ch,(xA5:bitstring,xA6:bitstring,xV3:bitstring,xT3:bitstring));
let r3=xor(xA5,h(con(xRIDi,xT3)) in
let r2=xor(xA6,h(con(xIDj,xT3))) in
let SKi=h(con(con(con(con(xIDj,xRIDi),r1),r2),r3)) in
let V3'=h(con(con(con(con(SKi,xIDj),xRIDi),r1),r3)) in
if V3'=xV3   then event UserAcDrone();0).
(* --------Server's process--------- *)
let UserReg=in(sch,(zIDi:bitstring));
new ki:bitstring;
new TIDi:bitstring;
new IDj:bitstring;
let RIDi=h(con(zIDi,ki)) in
let Ni=h(con(con(RIDi,ki),Ks)) in
let RIDi''=xor(RIDi,h(con(ki,Ks))) in
out(sch,(RIDi,Ni,IDj));0.
```

```
let DroneReg=in(sch,(zIDj:bitstring));
new kj:bitstring;
let Nj=h(con(con(zIDj,kj),Ks)) in
out(sch,(Nj));0.
let ServerAuth=in(ch,(zA1:bitstring,zA2:bitstring,zV1:bitstring,zTIDi:bitstring,zT1:bitstring));
new ki:bitstring;
new RIDi'':bitstring;
let RIDi=xor(RIDi'',h(con(ki,Ks))) in
let Ni=h(con(con(RIDi,ki),Ks)) in
let r1=xor(zA1,h(con(RIDi,zT1))) in
let IDj=xor(zA2,h(con(con(RIDi,Ni),zT1))) in
let V1'=h(con(con(con(RIDi,IDj),r1),zT1)) in
if V1'=zV1   then event ServerAcUser();
new r2:bitstring;
new T2:bitstring;
new kj:bitstring;
let Nj=h(con(con(IDj,kj),Ks)) in
let A3=xor(r2,h(con(IDj,Nj))) in
let A4=xor(RIDi,h(con(con(IDj,Nj),T2))) in
let V2=h(con(con(con(IDj,Nj),r2),T2)) in
out(ch,(zA1,A3,A4,V2,zT1,T2));0.
(* ------Drone's process------- *)
let ProcessDrone=new IDj:bitstring;
out(sch,(IDj));
in(sch,(yNj:bitstring));
!( in(ch,(yA1:bitstring,yA3:bitstring,yA4:bitstring,yV2:bitstring,yT1:bitstring,yT2:bitstring));
let RIDi=xor(yA4,h(con(con(IDj,yNj),yT2))) in
let r2=xor(yA3,h(con(IDj,yNj))) in
let V2'=h(con(con(con(IDj,yNj),r2),yT2)) in
if V2'=yV2   then event DroneAcServer();
new r3:bitstring;
new T3:bitstring;
let r1=xor(yA3,h(con(IDj,yNj))) in
let SKj=h(con(con(con(con(IDj,RIDi),r1),r2),r3)) in
let A5=xor(r3,h(con(RIDi,T3))) in
let A6=xor(r2,h(con(IDj,T3))) in
let V3=h(con(con(con(con(SKj,IDj),RIDi),r1),r3)) in
out(ch,(A5,A6,V3,T3));0).
let ProcessServer  = UserReg | DroneReg | ServerAuth.
(* --------- main---------- *)
process
          (!ProcessUser  |  !ProcessDrone | !ProcessServer  )
```

**Figure 7.** The process in the ProVerif tool.

```
Verification summary:
Query not attacker(SKi[]) is true.
Query not attacker(SKj[]) is true.
Query inj-event(UserStarted) ==> inj-event(UserAuthed) is true.
Query inj-event(ServerAcUser) ==> inj-event(DroneAcServer) is true.
Query inj-event(DroneAcServer) ==> inj-event(UserAcDrone) is true.
```

**Figure 8.** The results in the ProVerif tool.

*5.3. Informal Security Analysis*

5.3.1. Mutual Authentication

In this protocol, $U_i$ and $D_j$ realize mutual authentication with the help of $S$. $V_1$ in message $M_1$ is the authentication value for $S$ authenticating $U_i$, $V_2$ in message $M_2$ is the authentication value for $D_j$ authenticating $S$ and $V_3$ in message $M_3$ is the authentication value for $D_j$ authenticating $U_i$. Therefore, the proposed protocol realizes the mutual authentication between $U_i$ and $D_j$.

5.3.2. Replay Attacks

Our proposed protocol uses timestamps $T_1, T_2, T_3$. When $U_i, D_j$, or $S$ receive the message, it first verifies the freshness of the timestamp. If the timestamp is valid, the session process continues. When $A$ replays to a message transmitted from the common channel, the timestamp becomes invalid and the session process is terminated when an entity is verifying the timestamp. Thus, the proposed protocol can resist replay attacks.

5.3.3. Privileged Insider Attacks

If $A$ can obtain the long-term key $K_s$ of $S$, because $\{TID_i, RID_i^*, k_j\}$ and $\{ID_j, k_j\}$ stored in $S$ are unknown, $\{RID_i, ID_i, r_1, r_2, r_3\}$ cannot be computed and $A$ cannot compute $SK$, where $SK = h(RID_i \parallel ID_j \parallel r_1 \parallel r_2 \parallel r_3)$. If $A$ can obtain the parameter $\{TID_i, RID_i^*, k_j\}$ and $\{ID_j, k_j\}$ stored in $S$, $A$ can intercept message $M_1$ from the public channel, obtain $TID_i$ and then index it to $RID_i^*$, but the long-term key $K_s$ of $S$ is unknown, so $\{RID_i, ID_i, r_1, r_2, r_3\}$ cannot be computed. Thus, the proposed protocol can resist privileged insider attacks.

5.3.4. Drone Capture attacks

If $A$ can obtain the parameter $\{N_j\}$ stored in the drone's memory, the $RID_i$ cannot be computed because $A$ does not know the identity $ID_j$ of the $D_j$, where $RID_i = h(ID_j \parallel N_j \parallel T_2) \oplus A_4$. Furthermore, $A$ cannot compute $\{r_1, r_2, r_3\}$ and $SK$. Thus, the proposed protocol can resist drone capture attacks.

5.3.5. Man-in-the-Middle Attacks

Let us suppose that $A$ can intercept message $M_1 = \{A_1, A_2, V_1, TID_i, T_1\}$ transmitted on the public channel between $U_i$ and $S$. As $A$ cannot obtain the information $\{RID_i', ID_j', r\}$ in the smart card and $\{ID_i, PSW_i, BIO_i\}$ of $U_i$, $A$ cannot calculate the values $\{RID_i, ID_j, r_1\}$ required for $V_1$, where $V_1 = h(RID_i \parallel ID_j \parallel r_1 \parallel T_1)$. Therefore, after $A$ tampers with $M_1$, it cannot pass the authentication of $S$. Similarly, because the privacy value is unknown, $A$ cannot compute the value $V_2, V_3$ or $V_4$ and cannot complete the verification after intercepting the information $M_2, M_3$ or $M_4$. Therefore, the proposed protocol can resist man-in-the-middle attacks.

5.3.6. User Anonymity and Untraceability

The identities of $U_i$ and $D_j$ are not directly transmitted on the public channel and their identities cannot be computed. If $A$ wants to track $U_i$ or $D_j$, $A$ intercepts the message $\{M_1, M_2, M_3, M_4\}$ transmitted on the common channel, but the messages are variable during each session because random numbers $\{r_1, r_2, r_3\}$ are used. $A$ cannot track the $U_i$ or $D_j$. Therefore, the proposed protocol can provide user anonymity and untraceability.

**6. Security and Performance Comparisons**

In this section, we compare our protocol with those of Hussain et al. [21], Ever et al. [26], Wazid et al. [20] and Srinivas et al. [23] in terms of security, computational costs and communication costs.

### 6.1. Security Comparisons

In security comparison, ✓ indicates that the protocol can resist known attacks and × indicates that the protocol cannot resist attacks. The results of the security comparison are shown in Table 3. Here, in 2020, Ali et al. [41] found that the protocol of Srinivas et al. [23] could not provide anonymity and untraceability and was vulnerable to drone capture attacks. In the same year, Hussain et al. [21] pointed out that the protocol of Wazid et al. [20] could not realize mutual authentication and was vulnerable to privileged insider attacks and impersonation attacks. Deebak et al. [52] found that the protocol of Ever et al. [26] could not provide anonymity and untraceability and was vulnerable to drone capture attacks. We point out that the protocol of Hussain et al. [21] is vulnerable to drone capture attacks, privileged insider attacks and drone impersonation attacks in Section 3. So, we can see that proposed protocol can resist known attacks and has better security.

**Table 3.** Comparisons of security.

| Security Properties | [23] | [20] | [26] | [21] | Ours |
|---|---|---|---|---|---|
| Privileged insider attacks | − | × [21] | − | × | ✓ |
| Impersonation attacks | ✓ | × [21] | ✓ | × | ✓ |
| drone capture attacks | × [41] | ✓ | × [52] | × | ✓ |
| Mutual authentication | ✓ | × [21] | ✓ | ✓ | ✓ |
| User anonymity | × [41] | ✓ | × [52] | ✓ | ✓ |
| Perfect forword secrecy | − | − | − | ✓ | ✓ |
| Man-in-the-middle attacks | ✓ | ✓ | − | ✓ | ✓ |
| Temporary information disclose attacks | − | − | − | ✓ | ✓ |
| Untraceability | × [41] | ✓ | × [52] | ✓ | ✓ |

### 6.2. Performance Comparison

We compare the protocol with other related papers in terms of computational costs and communication costs. The computational costs includes the costs required to perform various operations during the login authentication process, because the computational costs of XOR and join operations are small enough to be ignored. Here, we performed a simulation experiment to evaluate the approximate computational time of the protocols. In the simulation experiment, we used Redmi note 9 Pro equipped with Android system, Qualcomm Snapdragon 750 processor and 8 G running memory to simulate users, using a Lenovo Desktop computer with Windows 10, Intel(R) Core(TM) i5-9500 CPU @ 3.00 GHz Processor and 8 G RAM to simulate servers. Since we had no suitable equipment to simulate drones, we used the results of Hussain et al. [21] in the simulation experiment as the computational time of drones. The experimental results are shown in Table 4. According to the experimental results, the fuzzy extraction function took the same time as the hash function, so we used the fuzzy extraction function as the hash function. The comparison of computational costs is shown in Table 5. We can see that the protocol of Srinivas et al. [23] and Wizard et al. [20] only use fuzzy extraction and hash operations. The computational costs of the proposed protocol is slightly higher than those of the above two protocols. The protocol of Ever et al. [26] uses elliptic curve scalar multiplication operation, bilinear pairing operation and hash operation. The protocol of Hussain et al. [21] uses symmetric encryption operation, fuzzy extraction operation and hash operation. Therefore, the computational costs of the protocol of Ever et al. [26] and Hussain et al. [21] are higher than those of other protocols.

**Table 4.** Experimental results.

| Operations | Symbolic | $U_i$ | $S$ | $D_j$ |
|---|---|---|---|---|
| Bilinear pairing | $T_{bp}$ | 38.9 ms | 9 ms | 12.52 ms |
| Symmetric encryption | $T_{se}$ | 0.0392 ms | 0.202 ms | 0.013 ms |
| Hash function | $T_h$ | 0.00251 ms | 0.0027 ms | 0.006 ms |
| Scalar multiplication | $T_{sm}$ | 20 ms | 9 ms | 4.107 ms |

**Table 5.** Computational cost comparison.

| Protocols | $U_i$ | $S$ | $D_j$ | Tocal | Tocal (ms) |
|---|---|---|---|---|---|
| Srinivas et al. [23] | $T_f + 14T_h$ | $9T_h$ | $9T_h$ | $T_f + 30T_h$ | 0.116 |
| Wazid et al. [20] | $T_f + 16T_h$ | $8T_h$ | $7T_h$ | $T_f + 31T_h$ | 0.106 |
| Ever et al. [26] | $2T_{bp} + 5T_h$ | $2T_{bp} + 3T_h$ | $4T_{sm} + 2T_{bp} + 9T_h$ | $4T_{sm} + 6T_{bp} + 17T_h$ | 137.34 |
| Hussain et al. [21] | $T_f + 15T_h$ | $2T_{se} + 9T_h$ | $7T_h$ | $2T_{se} + T_f + 31T_h$ | 0.510 |
| Ours | $T_f + 12T_h$ | $9T_h$ | $8T_h$ | $T_f + 29T_h$ | 0.135 |

Here, $T_{se}$ represents the time to perform the symmetric encryption operation, $T_{bp}$ represents the time to perform the the bilinear pairing operation, $T_{sm}$ represents the time to perform the elliptic curve scalar multiplication operation, $T_f$ represents the time to perform the fuzzy extraction function and $T_h$ represents the time to perform the hash operation.

In terms of communication costs, we compared the cost used to transmit messages on the common channel in the login authentication phase. Here, we assumed that the cost of transmitting the timestamp was 32 bits, the cost of transmitting identity and the random number was 160 bits, the cost of transmitting hash function was 256 bits and the cost of transmitting ECC points was 32 bits. Therefore, based on the above assumptions, we computed the communication cost of our protocol as an example. The computational methods of other protocols were similar. Our protocol transmitted three rounds of messages on the common channel, namely, $M_1 = \{A_1, A_2, V_1, TID_i, T_1\}$, $M_2 = \{A_1, A_3, A_4, V_2, T_1, T_2\}$ and $M_3 = \{A_5, A_6, V_3, T_3\}$. Among them, $\{V_1, V_2, V_3\}$ belonged to a hash value, $\{T_1, T_2, T_3\}$ belonged to the timestamp and $\{A_1, A_2, TID_i, A_3, A_4, A_5, A_6\}$ belonged to a random number. Therefore, the communication cost of our protocol was 2176 bits. Similarly, the communication costs of Srinivas et al. [23], Ever et al. [26], Wazid et al. [20] and Hussain et al. [21] were 1536 bits, 1696 bits, 5344 bits and 2061 bits, respectively. The comparison results of communication costs are shown in Table 6 and Figure 9 can more clearly describe the comparison results. It can be seen that the communication cost of the proposed protocol was much lower than that of the protocol of Ever et al. [26].

**Table 6.** Communication cost comparison.

| Protocols | Rounds | Communication Cost |
|---|---|---|
| Srinivas et al. [23] | 3 | 1536 bits |
| Wazid et al. [20] | 3 | 1696 bits |
| Ever et al. [26] | 6 | 5344 bits |
| Hussain et al. [21] | 3 | 2061 bits |
| Ours | 3 | 2176 bits |

According to the above comparison, it is clear that, in terms of security, our protocol can resist known attacks, whereas other protocols cannot resist known attacks. So, our protocol has better security than other protocols. In terms of computational costs, the proposed protocol is more expensive than the protocols of Srinivas et al. [23] and Wazid et al. [20] and has a lower computation cost than the protocols of Ever et al. [26] and Hussain et al. [21]. In terms of communication costs, although the proposed protocol is more expensive than the protocols of Srinivas et al. [23], Wazid et al. [20] and Hussain et al. [21], it has a much lower cost than the protocol of Ever et al. [26].

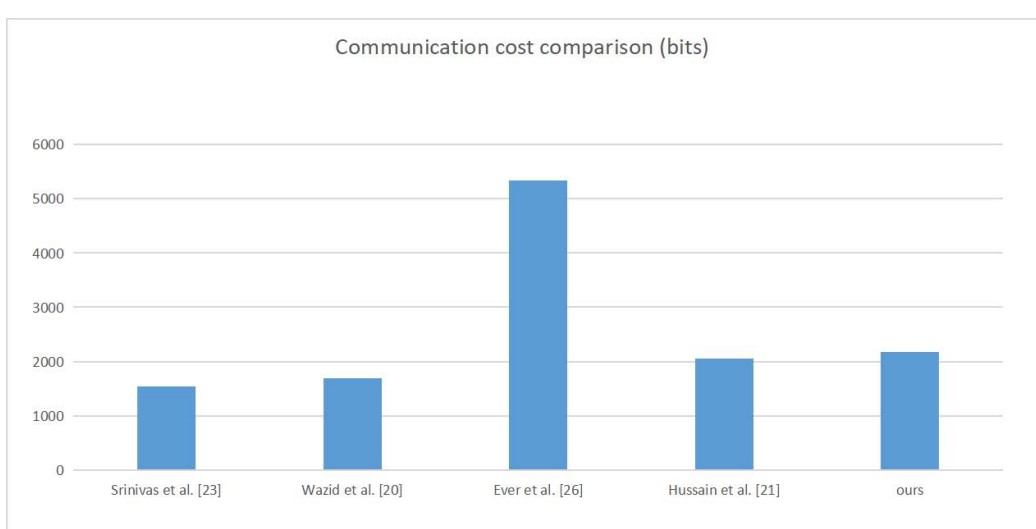

**Figure 9.** Communication cost comparison.

## 7. Conclusions

This paper first summarizes the importance and combination of *IoD* and 5*G*, reviews the recent AKA protocol in *IoD* and briefly reviews the protocol of Hussain et al. [21], pointing out that Hussain et al.'s protocol [21] is vulnerable to drone capture attacks, privileged insider attacks and session key disclosure attacks. To solve the security problems faced by the protocol of Hussain et al. [21], we propose an improved protocol. Through an informal analysis, we show that the proposed that protocol could resist known security attacks. In addition, the security and effectiveness of the protocol are demonstrated through a formal security analysis. Finally, through a comparison, we conclude that the protocol is secure compared with recent protocols. The rapid development of 5*G* makes the emergence of 6th generation mobile communication technology (6*G*) an inevitable trend and the subject of introducing 6*G* into *IoD* has a great research value in the future. In addition, researchers may combine drones with external smart devices to meet some specific needs. In future research work, it would also be necessary to design a secure authentication protocol for other architectures of drones. Therefore, the secure communication of *IoD* under different architectures is worthy of in-depth study by scholars.

**Author Contributions:** Conceptualization, S.K. and C.C.; methodology, T.W. and X.G.; software, Y.C.; formal analysis, X.G. and S.K.; investigation, Y.C. and C.C.; writing—original draft preparation, T.W. and X.G. All authors have read and agreed to the published version of the manuscript.

**Funding:** This research study received no external funding.

**Institutional Review Board Statement:** Not applicable.

**Informed Consent Statement:** Not applicable.

**Data Availability Statement:** Data are contained within the article.

**Conflicts of Interest:** The authors declare no conflict of interest.

## Abbreviations

*IoT*—Internet of Things; *IoD*—Internet of Drones; 5*G*—fifth-generation mobile communication technology; 4*G*—fourth-generation mobile communication technology; ROR—real-oracle random; AKA—authentication and key agreement; RFID—radio frequency identification; ECC—elliptic curve encryption; PKI—public key infrastructure; PUF—physical unclonable function; 6*G*—6th generation mobile communication technology.

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
