# Peer review of "Amassing the Security: An Enhanced Authentication Protocol for Drone Communications over 5G Networks"

_drones, doi:10.3390/drones6010010_

Round 1

Reviewer 1 Report

The authors consider in the work the important task of increasing the security of the involved drones from cyber attacks. The authors propose an improved protocol to improve security. Figure 1 shows the architecture under consideration, consisting of a control center, user, server and drones, in this regard, the question arises of how applicable this protocol is in a more encrypted architecture such as described in the work (https://ieeexplore.ieee.org/document/8795473,https://www.hindawi.com/journals/wcmc / 2021/6710074 /) when the architecture involves external smart devices with which the drones exchange data. 
In my opinion, this issue should be clarified in the paper.

Reviewer 2 Report

The authors show the insecurity of Hussain et al.’s protocol. Meanwhile, the authors put tremendous efforts to propose an enhanced protocol for drone communications over 5G networks. The security and comparative analysis of the proposed protocol with the existing benchmark shows better security features along with low communication and computation overheads. Overall, the paper is organized well and contributes to the existing authentication and key exchange. However, the following edits are required.

  1. Figure 1 needs more explanation.
  2. I suggest related work needs to be supported by tables. Meanwhile, the authors should discuss the limitations of current related work and how they overcame these limitations.
  3. In Section 5, the authors should provide simulation results to enhance reasonable analyses and discussions.

Reviewer 3 Report

The authors proposed an authentication protocol to secure drones from impersonation attack or when captured. The given methodology formally defined and the given notations can be understood well. I think the formalization of the methodology is sound but it requires some improvement as follows:

  1. The definition, process and queries in the ProVerif tool is given in Fig. 5~ 7. However, it is hard to understand the overview of the description. Therefore, it is better to illustrate the description using figures so that the mechanism can be understood.
  2. A theorem and proof was given in 4.1.2. The authors should define theorem distinguished from text. Please refer here to write Theorem in a proper way. 
  3. In Sect. 5.1, the security comparison was given. Table 2 shows the comparison but it is hard to follow why the comparison was given as shown in Table 2. The authors should elaborate more about the other protocol proposed by the literature. For example, why Srinivas et al. [8] method cannot provide anonymity and untraceability? Give the reason. This also applies to Ref. [15,19,16].
  4. In Sect. 5.2, performance comparison should also shows the lightweight performance showed by the proposed method. Communication cost was given in Fig. 9 but the proposed method cannot exceeds the performance of Srinivas and Wazid.  Related to this, computation cost are fewer as claimed in the manuscript. However, the author must provide the result for computation cost.

Round 2

Reviewer 1 Report

I recommend the manuscript for publication.